# Novel genetically encoded fluorescent probes enable real-time detection of potassium in vitro and in vivo

Helmut Bischof [1], Markus Rehberg[2], Sarah Stryeck[1], Katharina Artinger[3], Emrah Eroglu [1], Markus Waldeck-Weiermair[1], Benjamin Gottschalk[1], Rene Rost[1], Andras T. Deak[3], Tobias Niedrist[4], Nemanja Vujic[1], Hanna Lindermuth[5], Ruth Prassl [5], Brigitte Pelzmann[5], Klaus Groschner[5], Dagmar Kratky [1,6], Kathrin Eller[3], Alexander R. Rosenkranz[3], Tobias Madl [1,6], Nikolaus Plesnila [2], Wolfgang F. Graier [1,6] & Roland Malli[1,6]

Changes in intra- and extracellular potassium ion ($K^+$) concentrations control many important cellular processes and related biological functions. However, our current understanding of the spatiotemporal patterns of physiological and pathological $K^+$ changes is severely limited by the lack of practicable detection methods. We developed $K^+$-sensitive genetically encoded, Förster resonance energy transfer-(FRET) based probes, called GEPIIs, which enable quantitative real-time imaging of $K^+$ dynamics. GEPIIs as purified biosensors are suitable to directly and precisely quantify $K^+$ levels in different body fluids and cell growth media. GEPIIs expressed in cells enable time-lapse and real-time recordings of global and local intracellular $K^+$ signals. Hitherto unknown $Ca^{2+}$-triggered, organelle-specific $K^+$ changes were detected in pancreatic beta cells. Recombinant GEPIIs also enabled visualization of extracellular $K^+$ fluctuations in vivo with 2-photon microscopy. Therefore, GEPIIs are relevant for diverse $K^+$ assays and open new avenues for live-cell $K^+$ imaging.

[1] Institute of Molecular Biology and Biochemistry, Medical University of Graz, Neue Stiftingtalstraße 6/6, 8010 Graz, Austria. [2] Ludwig-Maximilians University (LMU), Institute for Stroke and Dementia Research (ISD), Klinikum der Universität München, Feodor-Lynen-Straße 17, 81377 Munich, Germany. [3] Clinical Division of Nephrology, Medical University of Graz, Auenbruggerplatz 27, 8036 Graz, Austria. [4] Clinical Institute of Medical and Chemical Laboratory Diagnostics, Medical University of Graz, Auenbruggerplatz 27, 8036 Graz, Austria. [5] Institute of Biophysics, Medical University of Graz, Neue Stiftingtalstraße 6/4, 8010 Graz, Austria. [6] BioTechMed-Graz, Graz, Austria. Correspondence and requests for materials should be addressed to R.M. (email: roland.malli@medunigraz.at)

Potassium ions ($K^+$), the most abundant intracellular cations[1], are essential for the proper functioning of all cell types[2]. Electrochemical $K^+$ gradients across the plasma membrane and membranes of organelles allow $K^+$ fluxes to control a variety of cell functions[3]. Disturbances of $K^+$ homeostasis have profound implications at both cellular and organismal level and feature in many diseases[1, 3] including neurological, cardio-vascular, renal, immunological, muscle, and metabolic disorders as well as cancer[4]. Besides its fundamental role in membrane potential, $K^+$ is also known to bind directly to several enzymes and regulate their activity, for example pyruvate kinase[5, 6], diol dehydratase[7], fructose 1,6-bisphosphatase[8], or

S-adenosylmethionine synthase[9]. Flux and transport of $K^+$ across bio-membranes occur via numerous different $K^+$ channels[10], exchangers[1], and pumps[11], which have emerged as promising drug targets for a variety of diseases[12]. However, our present understanding of extra- and intracellular $K^+$ fluctuations is very limited due to the lack of sensors that allow investigation of $K^+$ dynamics with high spatial and temporal resolution[13]. $K^+$-selective electrodes are often used to quantify $K^+$ in serum, plasma, or urine and to measure changes in extracellular $K^+$[14], but these electrodes are invasive and not able to measure spatiotemporal dynamics of $K^+$ variations and intracellular $K^+$ signals. Several small-molecule fluorescent $K^+$ sensors[15] have been developed

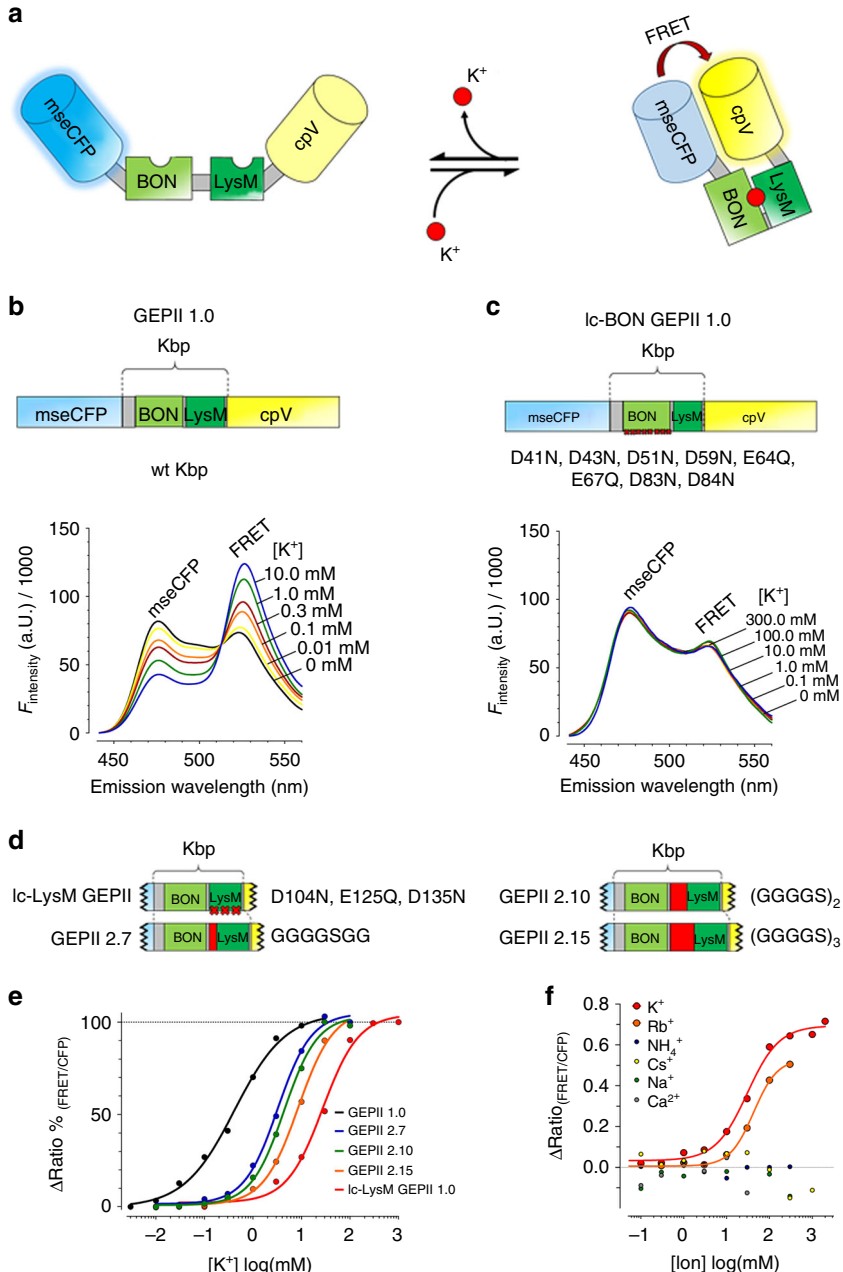

**Fig. 1** In vitro characterization of GEPIIs. **a** Schematic representation of the mechanism of $K^+$ sensing by FRET-based GEPIIs. Predicted conformational rearrangement of GEPIIs in the absence and presence of $K^+$. **b** Scheme (upper panel) and spectral properties (lower panel) of CFP and FRET fluorescence of purified GEPII 1.0 at increasing $K^+$ concentrations. Purified GEPII 1.0 was excited at $413.4 \pm 8$ nm. **c** Scheme (upper panel) and emission spectra (lower panel) of the $K^+$-insensitive lc-BON GEPII 1.0 in vitro. **d** Schematic overviews of differently mutated and engineered GEPIIs. **e** Normalized $EC_{50}$ curves for $K^+$ of GEPIIs in vitro ($n = 4$ for each construct). **f** $K^+$ selectivity of lc-LysM GEPII 1.0 ($n = 3$ for each ion). FRET ratio signals are plotted against increasing concentrations of different cations

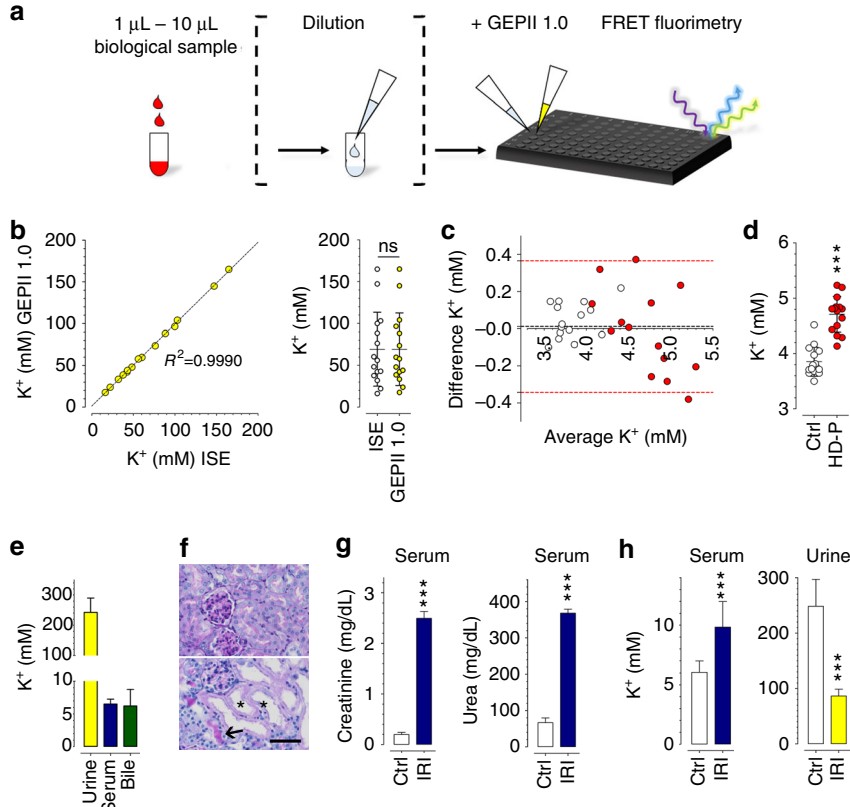

**Fig. 2** Application of purified GEPII 1.0 to determine K[+] levels within biological samples. **a** Schematic workflow of an automatable K[+] determination assay. **b** Urine [K[+]] of healthy human donors was determined using GEPII 1.0 in vitro and plotted against [K[+]] in the same samples determined with ion-selective electrodes (ISE, $n = 15$, $P = 0.9625$, paired $t$-test). **c** Bland–Altman plot showing human plasma [K[+]] of healthy donors (white circles, $n = 15$) and hemodialysis patients (red circles, $n = 15$) determined using ISE and purified GEPII 1.0, respectively. Bias (black dashed line) and 95% limits of agreement (red dashed lines) are shown ($n = 15$, $P = 0.6504$, paired $t$-test). **d** Comparison of individual K[+] levels of human plasma samples determined using recombinant GEPII 1.0 of healthy controls (ctrl, $n = 15 \pm$ SD) and hemodialysis patients (HD-P, $n = 15 \pm$ SD, ***$P < 0.0001$, unpaired $t$-test). **e** [K[+]] of mouse urine ($n = 11$, $\pm$SD), serum ($n = 25$, $\pm$SD), and bile ($n = 7$, $\pm$SD) samples quantified with recombinant GEPII 1.0. **f** Representative images of periodic acid-Schiff stained kidney slices of a control mouse (upper panel) and a mouse after ischemia-reperfusion injury (IRI). Scale bar represents 50 μm. Black arrow indicates a tubular cast; black stars mark dilated tubules with loss of brush borders. **g** Serum creatinine and serum urea levels of control mice (white bars) and mice after IRI (blue bars, $n = 5$ for each, $\pm$ SD, ***$P < 0.0001$, unpaired $t$-test). **h** Serum (left columns) and urine (right columns) [K[+]] of control mice (white bars, $n = 8$ for both $\pm$ SD) and mice after IRI (blue bar and yellow bar, $n = 18$ for serum, $n = 3$ for urine $\pm$ SD, ***$P < 0.001$, unpaired $t$-test)

with the goal of imaging K[+] fluctuations using fluorescence microscopy. Unfortunately, most of these fluorescent ionic indicators suffer from limited specificity for K[+] and low dynamic range, are difficult to load into cells, are not selectively targetable into subcellular compartments and may be toxic. Due to these severe restrictions, meaningful quantitative fluorescence K[+] imaging has been virtually impossible up to now[16]. Here we describe the development of a family of genetically encoded Förster resonance energy transfer- (FRET-) based K[+] indicators, which we have named GEPIIs (**G**enetically **E**ncoded **P**otassium **I**on **I**ndicators), and their validation for dynamic quantification of K[+] in vitro, in situ, and in vivo. We also present results which show that GEPIIs can be used successfully for K[+] fluorescence imaging, which will improve our understanding of (sub)cellular K[+] signals and K[+]-sensitive signaling pathways.

## Results

**Design and characterization of GEPIIs.** Very recently a bacterial K[+]-binding protein (Kbp), has been characterized[17]. Kbp consists of a K[+]-binding BON domain and a second lysine motif (LysM), which are supposed to interact in the presence of K[+][17]. We decided to explore whether Kbp could be used as the basis of a FRET-based K[+] probe, and fused either wild-type or mutated Kbp

directly with the optimized cyan and yellow FP variants[18], mseCFP and cpV, to the N- and C-terminus, respectively (Fig. 1). The mseCFP and cpV are approved FPs that have been used for the generation of many biosensors[19–22] due to their high FRET efficiency[18] and low tendency to form dimers[23]. We named these chimeras GEPIIs, as explained above, and hypothesized that upon K[+] binding to these chimeras, the two terminal FPs would be closely aligned yielding increased FRET, while in the absence of the ion, FPs would become separated resulting in reduced FRET (Fig. 1a). To test this idea, we first purified recombinant GEPII 1.0, containing wild-type Kbp (Fig. 1b, upper panel), and tested whether K[+] addition induced a fluorescence spectral change in vitro (Fig. 1b, lower panel). As expected, K[+] addition increased the FRET ratio signal of GEPII 1.0 (i.e., decrease of the FRET-donor mseCFP fluorescence accompanied by an increase in the FRET signal) in a concentration-dependent manner (Fig. 1b, e). The half maximal effective concentration (EC$_{50}$) of GEPII 1.0 was found to be 0.42 (0.37–0.47) mM of K[+] in vitro at room temperature (Fig. 1e). The response of the FRET ratio to K[+] covered a 3.2-fold range, which is remarkable high and should, hence, be sufficient for useful K[+] measurements. The high FRET ratio changes likely reflect a dramatic conformational rearrangement of Kbp from an elongated to a spherical structure upon K[+] binding which is in line with a recent report[17]. Values for the

association rate constant ($k_{on}$) and dissociation rate constant ($k_{off}$) of GEPII 1.0 were found to be $1.19 \times 10^{-1}$ mM$^{-1}$ s$^{-1}$ and $7.53 \times 10^{-1}$ s$^{-1}$ (Supplementary Fig. 1a), pointing to the fast on and off kinetics of the K$^+$ sensor. As expected, the absorption spectrum of GEPII 1.0 was unaffected by K$^+$ addition (Supplementary Fig. 1b).

Replacement of most acidic amino acids in the BON domain[17] by their corresponding amides yielded the K$^+$-insensitive "low-charge BON" (lc-BON) GEPII 1.0 (Fig. 1c). To generate a series of functional GEPIIs with lower K$^+$ sensitivity than GEPII 1.0, either all acidic amino acids within the LysM domain[17] were mutated or flexible linkers of variable lengths (7, 10, or 15 aa) were introduced between the BON and LysM domains (Fig. 1d and Supplementary Fig. 2). We hypothesized that in addition to classical mutations the introduction of variable linkers between the BON and LysM domain (Supplementary Fig. 2) will affect the K$^+$ sensitivity by impeding their K$^+$ dependent interaction. These rationally designed variants, which we named lc-LysM GEPII 1.0, GEPII 2.7, GEPII 2.10, and GEPII 2.15, sensed K$^+$ in vitro with EC$_{50}$ values of 27.43 (24.38–30.87) mM, 3.24 (2.96–3.55) mM, 4.39 (4.01–4.80) mM, and 8.59 (7.70–9.58) mM at room temperature, respectively (Fig. 1e, Supplementary Fig. 1c–f and Supplementary Tables 1 and 2). Analysis of the binding kinetics unveiled that all GEPIIs bind K$^+$ in a non-cooperative manner with Hill slopes close to 1 (Supplementary Fig. 3a–e), confirming that one molecule Kbp binds one K$^+$ ion[17]. In order to test the reversibility of purified GEPII 1.0, the recombinant sensor was immobilized using agarose. This approach allowed time-laps imaging of the purified K$^+$ probe in response to K$^+$ addition and removal and demonstrated both, the functionality and the reversibility of GEPII 1.0 in an agarose matrix (Supplementary Fig. 3f).

The K$^+$ sensitivity of GEPIIs significantly decreased with increasing temperature (Supplementary Fig. 4a). Taking into account the temperature dependence of the K$^+$ response, these probes enable the determination of K$^+$ concentrations ([K$^+$]) from $\geq 0.01$ mM up to $\leq 1000$ mM. This covers the entire range of known intra- and extracellular K$^+$ concentrations.

We then performed a series of experiments to further characterize the usefulness and robustness of the GEPII 1.0 K$^+$ signal. First, we used the temperature dependence of the response to examine the thermo-stability of recombinant GEPII 1.0. The FRET ratio signal of the purified probe was recorded repeatedly at 65 and 25 °C in the presence of K$^+$. On cooling, GEPII 1.0 repeatedly and completely recovered its K$^+$ sensitivity (Supplementary Fig. 4b). Next, we examined the pH dependence of the FRET ratio signal. The K$^+$ response of GEPII 1.0 was almost constant from pH 7.0 to 9.0 (Supplementary Fig. 4c), indicating that the probe can be used at normal intra- and extracellular pH values[24]. We then tested the ion selectivity of the high and low K$^+$-sensitive GEPII 1.0 and lc-LysM GEPII 1.0, separately, testing Na$^+$, Ca$^{2+}$, Cs$^+$, Rb$^+$, and ammonium ions (NH$_4^+$). The fluorescence properties of both probes remained virtually unaffected by Na$^+$ and Ca$^{2+}$ (Fig. 1f and Supplementary Fig. 4d). Only higher concentrations of Rb$^+$, Cs$^+$, and NH$_4^+$, which all have an atomic radius similar to K$^+$, moderately increased the FRET ratio signal of GEPII 1.0 (Supplementary Fig. 4d and Supplementary Table 1), while the fluorescence emission ratio of lc-LysM GEPII 1.0 was only influenced by Rb$^+$ but unaffected by Cs$^+$ and NH$_4^+$ (Fig. 1f and Supplementary Table 1). These experiments confirmed the high selectivity of GEPIIs for K$^+$ over Rb$^+$, Cs$^+$, NH$_4^+$, Na$^+$, and Ca$^{2+}$.

**Quantification of [K$^+$] in body fluids**. We investigated whether recombinant GEPII 1.0 could be used to quantify [K$^+$] in different biological samples in a fast, reliable, precise, and high-throughput manner using a conventional multi-well fluorescence plate reader.

We presumed that a GEPII-based K$^+$ assay could be accomplished in an automatable two- to three-step process including sample dilution, mixing with purified GEPII, and FRET fluorimetry (Fig. 2a). As a possible further simplification, we tested whether lyophilized GEPII 1.0 could be used. Lyophilized GEPII 1.0 was produced directly in a standard multi-well plate, stored for 1 week, reconstituted in aqueous solution in the presence of different K$^+$ levels, and analyzed using a fluorescence plate reader (Supplementary Fig. 5a). The results showed that recombinant GEPII 1.0 functionally fully recovered after freeze-drying and storage (Supplementary Fig. 5a), so that lyophilized GEPII 1.0 can be used for precise [K$^+$] measurement. An analytical calibration curve was established for the absolute quantification of [K$^+$] with purified GEPII 1.0 (Supplementary Fig. 5b). Next, we tested [K$^+$] measurement with GEPII 1.0 in human urine samples from healthy adults, compared to the results obtained with an ion-selective electrode (ISE), the current standard technology for clinical assays of electrolytes[14]. The results showed close agreement between the GEPII-based [K$^+$] quantification assay and the standard ISE (Fig. 2b and Supplementary Fig. 5c), demonstrating the accuracy and precision of the new method. Measurements of human plasma samples from healthy individuals and hemodialysis patients (HD-P) (Fig. 2c and Supplementary Fig. 5d), also agreed closely with ISE results, showing that GEPII 1.0 is functional, applicable, and precise in human plasma. As expected, plasma [K$^+$] was significantly higher in HD-P than in healthy controls (Fig. 2d). To test the stability of GEPII 1.0 in human urine and plasma, the same samples were analyzed immediately or 4 h after adding the probe to the samples. The values obtained were virtually identical after 4 h (Supplementary Fig. 5e), demonstrating the stability of the assay.

Due to the high K$^+$ sensitivity of GEPII 1.0, this assay requires only very small sample volumes (1–5 μl). To further test the practicability of the method with small samples, we then used GEPII 1.0 to determine [K$^+$] in single drops of murine blood, collected without the need to sacrifice the animals. Serum [K$^+$] of standard laboratory mice was found to be $6.52 \pm 0.76$ mM ($n = 25$, ±SD) with no significant difference between samples collected from the facial vein and the orbital sinus (Supplementary Fig. 6a). This value agrees well with published data[25]. We also determined [K$^+$] levels in urine and bile of individual mice using recombinant GEPII 1.0 (Fig. 2e), showing that the probe can be used in different body fluids with variable [K$^+$]. The GEPII 1.0 probe also proved stable in murine serum and urine samples for 4 h (Supplementary Fig. 6b). To further challenge and exploit the GEPII-technology we used a mouse model of kidney dysfunction, using surgically inflicted renal ischemia-reperfusion injury (IRI)[26]. Histological analysis revealed clearly pathological structures of tubular casts with a loss of brush borders and cell nuclei—features of severe renal injury[26]—in comparison to controls (Fig. 2f and Supplementary Fig. 6c). Classical clinical parameters of renal function such as serum creatinine and serum urea levels (Fig. 2g) as well as serum neutrophil gelatinase-associated lipocalin (Supplementary Fig. 6d) were also increased, demonstrating that the method did in fact cause manifest kidney injury[26]. We then applied the GEPII-technology to quantify [K$^+$] in serum and urine of mice with IRI and controls. These measurements clearly revealed increased serum and decreased urine [K$^+$] in mice with IRI compared to controls (Fig. 2h). Importantly, the samples did not show any differences in hemolysis (Supplementary Fig. 6e), so that the increased [K$^+$] in mice with IRI can be attributed to kidney dysfunction rather than release of K$^+$ from erythrocytes.

**Time-lapse [K$^+$]$_{ex}$ determination as cell viability assays**. Vital cells maintain a steep [K$^+$] gradient across the plasma membrane,

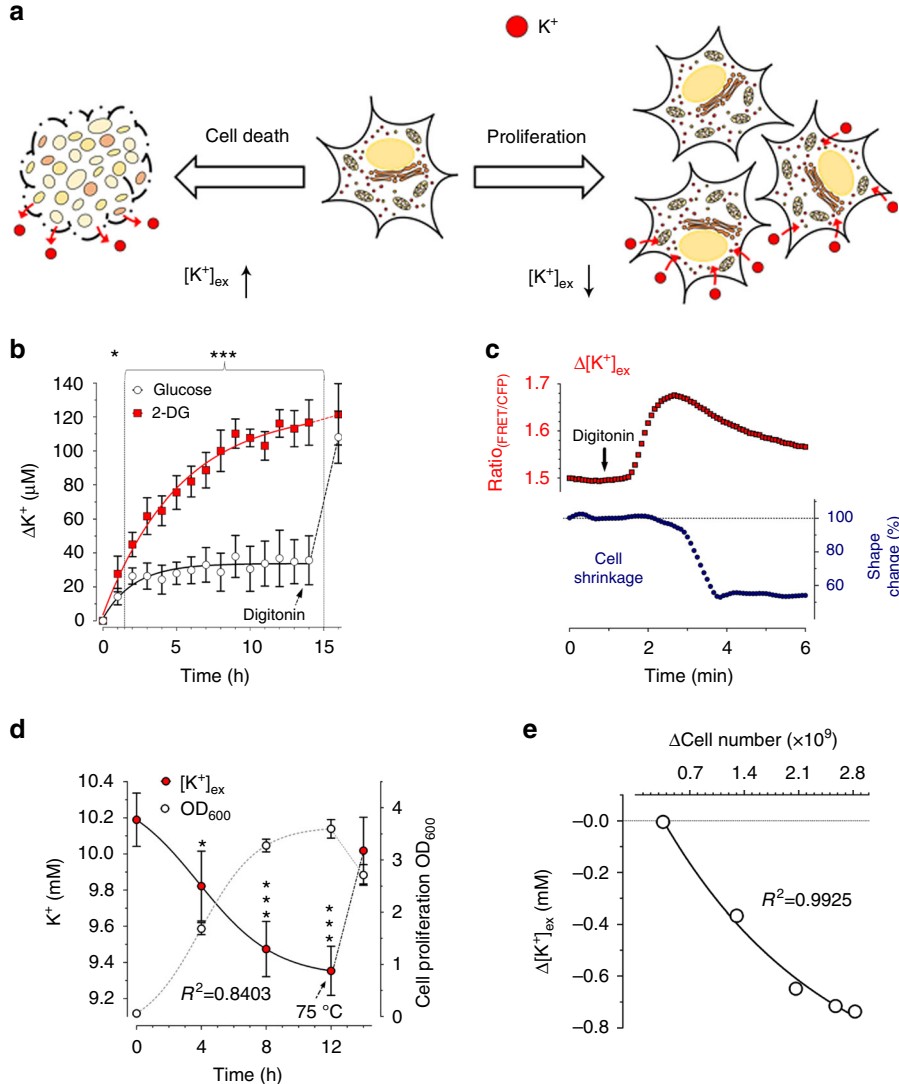

**Fig. 3** Application of purified GEPII 1.0 to determine cell viability and cell growth. **a** Scheme demonstrating cellular $K^+$ loss upon cell death (left arrow) or $K^+$ uptake of proliferating cells (right arrow). **b** $[K^+]$ over time determined with GEPII 1.0 within the supernatant of INS-1 cells cultured either in the presence of 10 mM glucose (white circles and black line, $n = 6 \pm$ SD) or 10 mM 2-deoxyglucose (red squares and line, $n = 6 \pm$ SD). As indicated, 50 μM digitonin was applied after 14 h. $*P = 0.0178$, $***P < 0.0005$, unpaired $t$-test. **c** Representative FRET ratio signals over time of recombinant GEPII 1.0 within the supernatant of HeLa cells (upper red curve) were recorded simultaneously with morphological alterations (cell shrinkage) of cells (lower blue curve) on an inverted fluorescence microscope. As indicated, 30 μM digitonin was added to induce cell necrosis. **d** $[K^+]$ of cell culture medium over time determined with purified GEPII 1.0 (red circles, $n = 5 \pm$ SD, $*P = 0.05$, $***P < 0.005$, one-way ANOVA test with Tukey's Multiple Comparison Test) during bacterial (*E. coli*) proliferation. Cell density was recorded in parallel by measuring the optical density at 600 nm ($OD_{600}$, white circles, $n = 5 \pm$ SD). As indicated $[K^+]_{ex}$ and $OD_{600}$ were determined after heating of the cells (75 °C for 1 h). **e** Increases of bacterial cell numbers (ΔCell number) were plotted against the respective decreases of the extracellular $K^+$ concentrations ($\Delta[K^+]_{ex}$)

with a ratio of intracellular to extracellular $[K^+]$ ($[K^+]_i$: $[K^+]_{ex}$) of $\geq 20$[1]. Since this gradient is maintained mainly via the sodium–potassium adenosine triphosphatase ($Na^+/K^+$-ATPase)[11], which is strictly energy dependent[27], defects in cellular energy metabolism, are generally accompanied by a loss of this $K^+$ gradient[28], with a rise of $[K^+]_{ex}$ (Fig. 3a, left); a situation that often precedes cell death[28]. Vice versa, in a closed system such as a cell culture, it can be expected that as cells proliferate they capture increasing amounts of $K^+$ so that the $[K^+]$ concentration in the cell culture medium ($[K^+]_{ex}$) decreases in proportion to the cell mass[1] (Fig. 3a, right). Monitoring $[K^+]$ in cell culture media could be a useful indicator of dynamic cell viability, cell death, and cell proliferation. We therefore tested whether recombinant GEPII 1.0 could be used to follow $[K^+]$ in cultures over periods of several hours. We did this first with INS-1 cells, a pancreatic beta-cell

line[29]. Cells were kept in the presence of glucose or its cytotoxic antimetabolite 2-deoxyglucose[30] (2-DG) and $[K^+]_{ex}$ was measured over time using purified GEPII 1.0 (Fig. 3b). While $[K^+]_{ex}$ remained almost constant in the presence of glucose, it gradually increased in the presence of 2-DG (Fig. 3b), pointing to the breakdown of the cellular $K^+$ gradient, an expected consequence of the severe cellular energy crisis and cell death caused by the cytotoxic glycolysis inhibitor[30]. At the end of this experiment cells were treated with digitonin, which increased $[K^+]_{ex}$ in the medium of control cells to similar levels as obtained by 2-DG treatment (Fig. 3b), confirming the presence of the same cell numbers under both conditions. If we assume that all cells of the same type have a similar $[K^+]_i$[1], the GEPII-based $[K^+]_{ex}$ assay with permeabilized cells could be used to estimate total cell numbers. We investigated the correlation between cell number and $[K^+]_{ex}$ increase upon cell

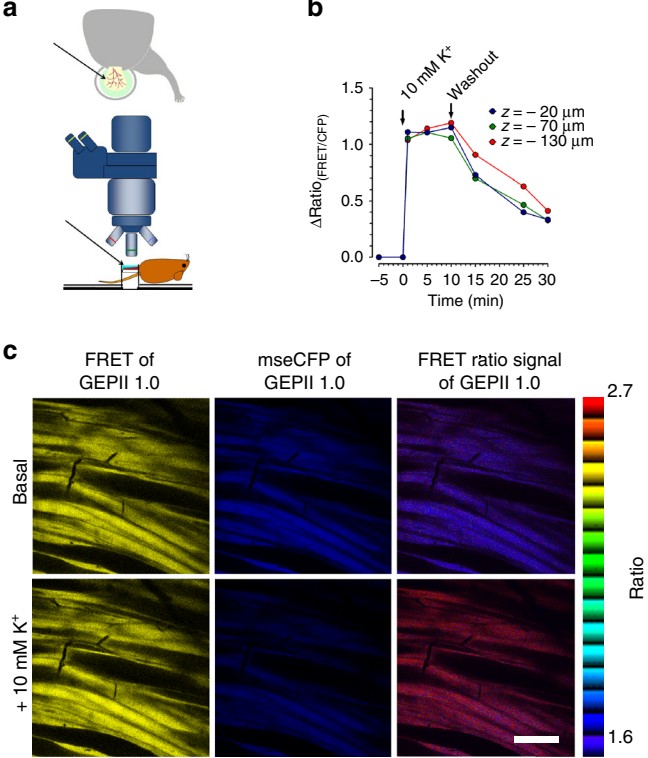

**Fig. 4** Testing the functionality of recombinant GEPII 1.0 in vivo. **a** Schemes showing the area of interest (musculus cremaster, arrow, upper panel) and the mouse prepared for imaging using the 2-photon imaging device (lower panel). **b** Representative in vivo FRET ratio signals of GEPII 1.0 loaded into the cremaster tissue of a living mouse. Signals were imaged using a 2-photon fluorescence imaging system at different $z$-positions. As indicated 10 mM KCl was applied to the tissue during intravital microscopy and subsequently washed out using a perfusion system. **c** Representative images showing the FRET signal (yellow, left images), mseCFP signal (blue, middle images) and FRET ratio signal (pseudocolored, right images) of GEPII 1.0 loaded into the musculus cremaster prior to (upper images) and upon the addition of 10 mM KCl (lower images) as indicated in **b**. White scale bar in the lower right image represents 100 µm

permeabilization, and, as expected, found a strict linear correlation between both parameters (Supplementary Fig. 7a).

To further demonstrate the broad applicability of purified GEPII 1.0, morphological alterations of necrotic cells and $[K^+]_{ex}$ were co-imaged using a conventional fluorescence wide-field microscope (Supplementary Fig. 7c). The highly sensitive GEPII 1.0 was suitable to dynamically quantify rises of $[K^+]_{ex}$ prior to detectable changes in the cell morphology of HeLa cells getting necrotic (Fig. 3c and Supplementary Movie 1). Similar results were obtained using other cell types, while the amplitude and plateau of the $[K^+]_{ex}$ signal correlated with the cell confluence (Supplementary Fig. 7b, c and Supplementary Movie 2). Furthermore, we used GEPII 1.0 to quantify the reduction of $[K^+]_{ex}$ during bacterial cell proliferation in simple shake flasks. Our data unveiled that during the exponential phase of bacterial growth $[K^+]_{ex}$ gradually decreased from 10.2 mM to 9.4 mM within the culture media (Fig. 3d). Interestingly, these experiments showed the correlation between increasing bacterial cell number and decreasing $[K^+]_{ex}$ (Fig. 3e), a relationship, which might be highly cell-type specific and relevant for optimizing bacteria growth conditions. Heating of the bacteria to release the intracellular $K^+$ increased $[K^+]_{ex}$ back to $\geq 10$ mM (Fig. 3d) and resulted in positive PI-staining (Supplementary Fig. 7d), indicating

pronounced permeabilization. Short-time treatment of bacteria with ampicillin also increased PI fluorescence in contrast to other antibiotics (Supplementary Fig. 7d). However, all antibiotics tested yielded in significant bacterial $K^+$ release, which could be detected with GEPII 1.0 (Supplementary Fig. 7e). Microorganisms are typically cultured on agar plates. Thus, we tested the functionality of GEPII 1.0 under such conditions. GEPII 1.0 embedded into agar–agar was able to report different predefined (Supplementary Fig. 7f) as well as dynamically manipulated $[K^+]$ (Supplementary Fig. 7g) in the gelatinous matrix. These experiments demonstrate that recombinant GEPIIs are suitable to visualize $K^+$ alterations both, in liquid media and agar.

**Testing the functionality of GEPII 1.0 in vivo.** To test the suitability of GEPIIs in vivo, recombinant GEPII 1.0 was loaded into the exposed striated muscle (musculus cremaster) of a mouse (Fig. 4a). Tissue loading was achieved by incubation with GEPII 1.0 for 20 min. Intravital microscopy showed that GEPII 1.0 diffused deep into the cremaster muscle without entering blood vessels, accumulating in the interstitial spaces (Supplementary Fig. 8). The fluorescence signals remained stable in this experimental setup, indicating that the recombinant protein probe was stable when applied in vivo. To test the functionality of GEPII 1.0 in the muscle, $[K^+]_{ex}$ was manipulated by the addition and removal of KCl (Fig. 4b, c). Increasing $[K^+]$ significantly increased the FRET ratio signal of recombinant GEPII 1.0 at different tissue depths, while the washout of $K^+$ reduced the signals (Fig. 4b). The FRET ratio decrease upon $K^+$ washout was slower in deeper tissue (Fig. 4b) and faster in more superficial tissue. These results demonstrate the feasibility of using GEPIIs to monitor spatial and temporal patterns of $[K^+]_{ex}$ changes in vivo.

**Quantification of intracellular $[K^+]$ of living cells.** The other major use of GEPIIs we wanted to test was monitoring of $K^+$ inside cells by microscopic imaging. We did this using cells expressing GEPIIs and manipulated $[K^+]_i$ by treating the cells with digitonin or gramicidin to permeabilize the plasma membrane. The FRET ratio signal of GEPIIs gradually decreased in the absence of extracellular $K^+$ (Supplementary Fig. 9a), reflecting initially high cytosolic $[K^+]$ being dissipated by these agents. Permeabilized cells were then treated with different $[K^+]_{ex}$, which dynamically affected mseCFP (FRET donor) and FRET acceptor fluorescence intensities (Fig. 5a, b); this showed that that GEPIIs indicate changes of $[K^+]_i$ reversibly and in a ratiometric manner. The respective changes of FRET ratio signals were induced by adding and removing $K^+$ either in a cumulative (Fig. 5b and Supplementary Fig. 9b) or repetitive manner (Supplementary Fig. 9c) to determine $EC_{50}$ values and $K^+$ selectivity of GEPIIs in cells (Fig. 5c and Supplementary Fig. 9d, Supplementary Table 2). As under physiological conditions $K^+$ and $Na^+$ are the predominant ions[1, 6], we next investigated whether or not the presence of $Na^+$ at concentrations ranging from 0 mM to 150 mM impact on $K^+$ sensing. Even strong $Na^+$ fluctuations did not significantly influence the FRET ratio signals of GEPII 1.0 in response to $K^+$ of permeabilized HeLa cells (Supplementary Fig. 10).

We then investigated whether the FRET-based $K^+$ sensors could detect possible variations of $[K^+]_i$ between distinct cellular compartments. To do this, GEPIIs of different $K^+$ sensitivities were targeted to various organelles and cellular subdomains (Supplementary Fig. 11a) using approved targeting sequences and basal FRET ratio signals were recorded in intact cells (Fig. 5d). Average FRET ratio signals of the $K^+$-insensitive lc-BON GEPII 1.0 were not affected by targeting (Supplementary Fig. 11b, c). In contrast, FRET ratios of the $K^+$-sensitive GEPII 1.0

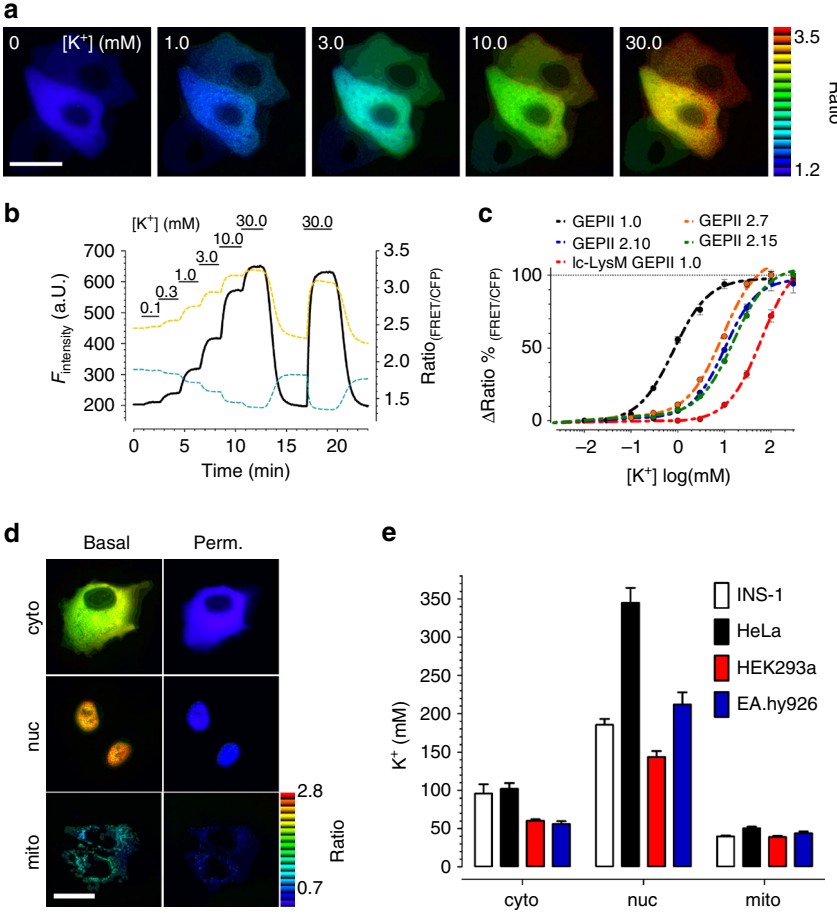

**Fig. 5** Characterization and application of GEPIIs in cells. **a** Representative pseudocolor ratio images of permeabilized HeLa cells expressing GEPII 1.0 in the presence of different [K$^+$]. Scale bar represents 20 μm. **b** CFP (cyan dashed line), FRET (yellow dashed line) fluorescence intensities, and the respective FRET ratio signal (black solid line) of GEPII 1.0 expressed in HeLa cells. Permeabilized (5 μM digitonin) HeLa cells were treated with different [K$^+$] as indicated using a semi-automatic perfusion system. **c** Normalized EC$_{50}$ curves for K$^+$ of different GEPIIs expressed in HeLa cells ($n = 6$ for each ± SD). Cells were permeabilized using 5 μM digitonin and respective K$^+$ concentrations were added. Respective ΔFRET ratio values, normalized to the maximal response, i.e., 100%, were plotted against [K$^+$]. **d** Representative ratio images of HeLa cells expressing either cytosolic (upper panels), nuclear (middle panels) or mitochondrially targeted (lower panels) lc-LysM GEPII 1.0 under resting conditions (intact cells, left panels) or after cell permeabilization (10 μM digitonin) in the absence of extracellular K$^+$ (right panels). Scale bar represents 20 μm. **e** Basal [K$^+$] ± SEM calculated using lc-LysM GEPII 1.0 expressed in INS-1 (white bars), HeLa (black bars), HEK293a (red bars) and EA.hy926 cells (blue bars) within the cytosol ($n = 7$ independent experiments for each/$n =$ 52 cells for INS-1/ $n = 74$ cells for HeLa/$n = 62$ cells for HEK293a/$n = 29$ cells for EA.hy926), nucleus ($n = 7$ independent experiments for each/$n = 133$ cells for INS-1/$n = 35$ cells for HeLa/$n = 184$ cells for HEK293a/$n = 49$ cells for EA.hy.926), or mitochondria ($n = 7$ independent experiments for each/$n =$ 142 cells for INS-1/$n = 62$ cells for HeLa/$n = 102$ cells for HEK293a/$n = 45$ cells for EA.hy926)

(Supplementary Fig. 11b, c) and lc-LysM GEPII 1.0 showed organelle-specific differences with highest levels in the nucleus and low levels within mitochondria (Fig. 5d and Supplementary Fig. 11c). The differences of FRET ratio values between the cytosol and nucleus were also observed in single cells expressing non-targeted lc-LysM GEPII 1.0, which was equally distributed within both compartments (Supplementary Fig. 11a, d). The differences were equalized by membrane permeabilization (Fig. 5d and Supplementary Fig. 11d). Under consideration that targeting of GEPIIs does not cause any significant artefacts, quantifications of [K$^+$]$_i$ in the different subcellular compartments revealed that in diverse cell types the nuclear K$^+$ ([K$^+$]$_{nuc}$) is always significantly higher than the cytosolic K$^+$ concentration ([K$^+$]$_{cyto}$) (Fig. 5e) with [K$^+$]$_{nuc}$/[K$^+$]$_{cyto}$ ratios of 3.38, 1.94, 2.39, and 3.77 in HeLa, INS-1, HEK293a, and EA.hy926 cells, respectively. In contrast, targeted lc-LysM GEPII 1.0 revealed lower K$^+$ concentrations within mitochondria ([K$^+$]$_{mito}$) compared to [K$^+$]$_{cyto}$ of living cells (Fig. 5e) with [K$^+$]$_{mito}$/[K$^+$]$_{cyto}$ ratios of 0.49, 0.42, 0.65, and 0.78 in HeLa, INS-1, HEK293a, and EA.hy926 cells, respectively.

**Real-time visualization of intracellular K$^+$ signals**. To trigger intracellular K$^+$ alterations, we depolarized excitable INS-1 cells and monitored [K$^+$]$_{cyto}$ simultaneously with cytosolic Ca$^{2+}$ concentration ([Ca$^{2+}$]$_{cyto}$) using lc-LysM GEPII 1.0 in combination with CAR-GECO1, a far-red fluorescent genetically encoded Ca$^{2+}$ sensor (Fig. 6a). As already observed[31, 32], [Ca$^{2+}$]$_{cyto}$ transiently increased in response to cell depolarization due to the activation of voltage-gated Ca$^{2+}$ channels. With a short delay after Ca$^{2+}$ entry, the lc-LysM GEPII 1.0 signal clearly indicated a transient reduction of [K$^+$]$_{cyto}$ (Fig. 6a and Supplementary Fig. 12a). Importantly, the FRET ratio signal of the high-sensitivity GEPII 1.0 (Supplementary Fig. 12b) as well as the K$^+$-insensitive lc-BON GEPII 1.0 (Supplementary Fig. 12c) remained unaffected under these conditions. These results confirm that lc-LysM GEPII 1.0 had indeed detected a change in intracellular K$^+$ in real time. A similar K$^+$ signal was also observed upon depolarization of MIN6 cells (Supplementary Fig. 12d), another rodent pancreatic beta cell line[29]. To further characterize the transient cytosolic K$^+$ decrease, INS-1 cells were treated with tetraethylammonium

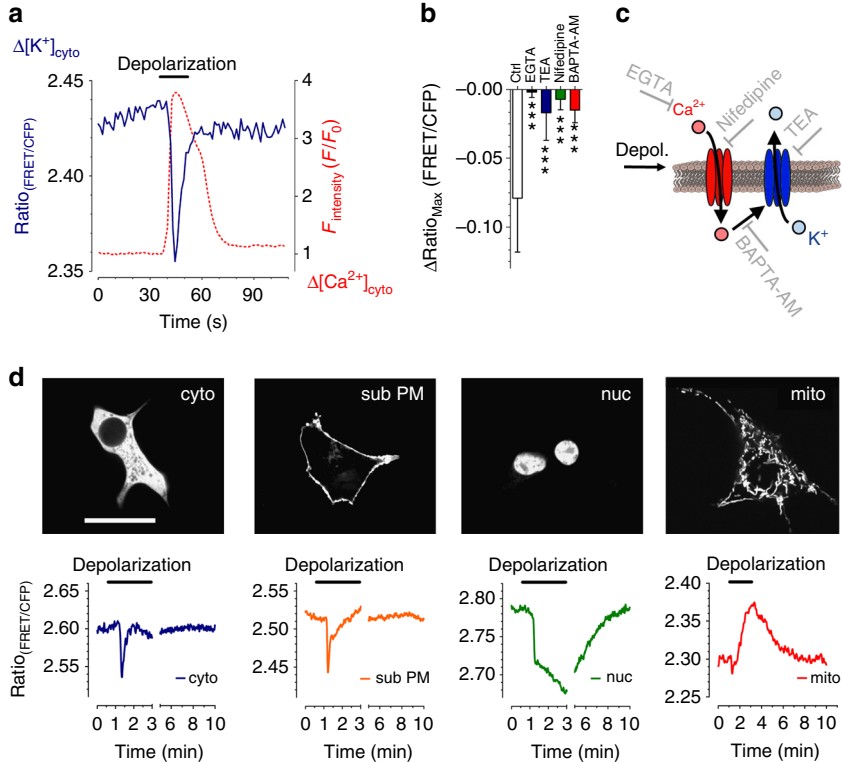

**Fig. 6** Real-time imaging of intracellular $K^+$ fluxes. **a** Representative single cell $K^+$ (blue solid line) and $Ca^{2+}$ (red dashed line) response of an INS-1 cell expressing lc-LysM GEPII 1.0 and CAR-GECO1 upon cell depolarization using 70 mM KCl ($n = 9$ independent measurements/32 cells/32 cells responded as demonstrated). **b** Columns represent maximal $\Delta$FRET ratio signals $\pm$ SD of INS-1 cells expressing cytosolic lc-LysM GEPII 1.0 upon depolarization under control conditions (ctrl, white bar, $n = 17/96$), in the absence of extracellular $Ca^{2+}$ (1 mM EGTA, black bar, $n = 7/50$), in the presence of 15 mM TEA (blue bar, $n = 6/50$), in the presence of 100 µM nifedipine (green bar, $n = 6/41$), and in cells loaded with BAPTA-AM (red bar, $n = 6/53$). ***$P < 0.0001$, one-way ANOVA test with Tukey's Multiple Comparison Test. **c** Schemes demonstrating the activation of $K^+$ efflux via a plasma membrane $K^+$ channel by $Ca^{2+}$ entry and points of action of pharmacological inhibitors. **d** Representative confocal images of cytosolic, subplasmalemmal, nuclear and mitochondrially targeted lc-LysM GEPII 1.0 in INS-1 cells (upper images). Scale bar represents 20 µm. Representative single cell responses of cytosolic ($n = 7/53/53$), subplasmalemmal ($n = 7/56/56$), nuclear ($n = 7/30/23$), and mitochondrially ($n = 7/45/33$) targeted lc-LysM GEPII 1.0 upon cell depolarization using 70 mM KCl (lower panels)

chloride (TEA), a frequently used $K^+$-channel blocker[33], prior to cell depolarization. When the cells were incubated with TEA, the FRET ratio signal of cytosolic lc-LysM GEPII 1.0 gradually increased, a sign that inhibition of plasma membrane $K^+$ channels induces accumulation of $[K^+]_{cyto}$ (Supplementary Fig. 12e). TEA did not affect the $Ca^{2+}$ signal upon depolarization (Supplementary Fig. 12e), but the transient drop in $[K^+]_{cyto}$ was strongly or completely inhibited (Fig. 6b and Supplementary Fig. 12e).

We next tested the effects of inhibiting the cytosolic $Ca^{2+}$ signal. This was done done in different ways: chelating extracellular $Ca^{2+}$ with EGTA (Fig. 6b and Supplementary Fig. 12f), blocking voltage-gated $Ca^{2+}$ channels with nifedipine (Fig. 6b), and buffering intracellular $Ca^{2+}$ elevations by BAPTA-AM loading (Fig. 6b and Supplementary Fig. 12g). All of these treatments considerably reduced the intracellular $K^+$ transient in response to cell depolarization. Taken together, these experiments show that the GEPII $K^+$ signal follows exactly the expected pattern for the usual depolarization mechanism, in which $Ca^{2+}$ entry via voltage-gated channels evokes activation of $K^+$ channels, followed by $K^+$ efflux from the cells (Fig. 6c).

Finally, we investigated whether targeted lc-LysM GEPII 1.0 (Fig. 6d, upper panel) could be used to monitor subcellular $K^+$ dynamics during cell depolarization. Despite almost identical transient $K^+$ signals in the cytosol and subplasmalemmal area, $[K^+]_{nuc}$ and $[K^+]_{mito}$ were dynamically affected in a completely different manner (Fig. 6d, lower panel). Though $[K^+]_{nuc}$ was higher than $[K^+]_{cyto}$, it exhibited a sustained reduction during

stimulation, while the low $[K^+]_{mito}$ gradually increased in response to cell depolarization (Fig. 6d). In this way, live-cell imaging with targeted GEPIIs was able to detect the dynamic organelle-specific $K^+$ pools that are differently reshuffled in response to cell activation.

## Discussion

Based on Kbp we developed novel protein-based FRET probes, the GEPIIs, that enable dynamic quantification of biologically relevant $K^+$ variations in vitro, in situ, and in vivo. Both, mutations of all acidic amino acids within the wild-type LysM domain and the introduction of flexible linkers between the BON and LysM domain of Kbp significantly decreased the $K^+$ affinity of respective chimeras. While these rationally designed constructs emerged suitable to image intra and extracellular $K^+$ fluctuations, other strategies might further improve the characteristics of Kbp-based GEPIIs in future. Already mutations of single acidic amino acids within either the BON or LysM domain might significantly alter the $K^+$ affinity of the probe. In analogy to single FP-based $Ca^{2+}$ probes, the GECOs[34], error-prone PCR, and site-directed evolution approaches also have the potency to generate improved GEPII variants. However, being already able to characterize spatial and temporal changes of $[K^+]$ with the available GEPIIs is a key to better understanding of $K^+$-based signaling[13], which is so important in physiology[1] and pathology[3]. Our data demonstrate that recombinant GEPIIs can measure $K^+$ levels in

different body fluids as precisely as ISE[14]. This can be done conveniently in a conventional multi-well fluorescence plate reader and has the advantage of a very small sample volume (around 5 μl) compared to common ISE instruments and most other $K^+$ indicators[15, 16]. This could enable more repetitive sampling from small animals without sacrificing them[25], and could also be useful in neonatal diagnostics[35] and forensic medicine[36]. We also demonstrated that an increase of $[K^+]_{ex}$ upon cell death is dynamically detectable in vitro and in situ with the GEPII-technology. Also, uptake of $K^+$ by proliferating bacteria can be determined over time as a dynamic measure of cell viability and proliferation rates. These findings are in line with other reports that point to the importance of $[K^+]_{ex}$ for cell growth[17, 37].

Having established that GEPIIs can measure $K^+$ accurately in solution and in biological samples, we went on to explore their usefulness in imaging applications. Intravital microscopy in the striated muscle preparation of living mice showed that GEPIIs can be used to follow spatiotemporal changes in $[K^+]_{ex}$ in vivo. These results suggest that GEPIIs could be used for dynamic imaging of $K^+$ fluxes in the brain, similar to the methods already established for $Ca^{2+}$. However, calcium ions are not directly involved in the generation of neuronal action potentials[38]; being able to directly monitor $[K^+]_{ex}$ in the living brain could be a great step forward in our ability to study neuronal activity[39].

We also applied GEPIIs to intracellular distribution of $K^+$ in organelles and subdomains of living cells using approved transfection protocols. These experiments not only confirmed the feasibility of the method but also immediately yielded the interesting results that $[K^+]$ is higher in the nucleus and lower in the mitochondria than in the cytosol of several different cell types. However, organelle targeting of fluorescent probes can cause artefacts due to differences in auto fluorescence, viscosity, and composition, which might produce false results. To exclude any of these disturbances, we tested three different GEPIIs with same FPs but different $K^+$ affinities within the different cellular compartments. Nevertheless the high $[K^+]_{nuc}$ is surprising, as until now there has been little evidence on the distribution of $K^+$ within organelles, apart from studies from the 1970s[40] and 1980s[41] that indicated the existence of $K^+$ gradients between the cytosol and nucleus. $K^+$ within the nucleus has been suggested to contribute to the control of gene expression[41], but although some $K^+$ channels and transporters are known to be present in the nuclear envelope[13, 42], the mechanisms that might dynamically regulate $[K^+]_{nuc}$[42] and $[K^+]_{mito}$[42, 43] are not well understood. Although nuclear pores allow the passage of macromolecules[44], their permeability for $K^+$ ions remains elusive. Several studies[45–47] demonstrated that certain DNA structures specifically bind $K^+$ with high affinity, indicating a certain role of $K^+$ within the nucleus. However, further experiments are necessary to validate any peculiarities of the $K^+$ homeostasis of cellular nuclei.

By the usage of targeted GEPIIs, we are now also able to visualize $K^+$ accumulation within mitochondria in response to $Ca^{2+}$ elevation in individual cells in real time. Our findings confirm the assumption of concentration and electrical gradients that facilitate $K^+$ uptake by energized mitochondria[42, 48]. These imaging experiments also tend to confirm the existence of $Ca^{2+}$-activated $K^+$ channels in the inner mitochondrial membrane and reveal dynamic $K^+$ changes in the organelle in pancreatic beta cells. $K^+$ imaging with GEPIIs will allow testing of diverse activators, inhibitors, and modulators of plasma membrane as well as intracellular $K^+$ channels in a practicable, high-throughput and high-content manner. In conclusion, our results show that GEPIIs expressed in living cells are a practical tool for further investigating subcellular $K^+$ dynamics, the mechanisms that control them, and the consequences for cell signaling events and related biological functions.

## Methods

**Chemicals and buffer solutions**. Materials for cell culture were purchased from PAA laboratories (Pasching, Austria). Restriction enzymes, chemically competent 10-beta *Escherichia coli* cells for cloning and chemically competent *E. coli* BL21 (DE3) cells for protein expression were obtained from New England Biolabs (Ipswich, MA, USA). Agar–Agar Kobe I, $CaCl_2$, $CsCl$, D-Glucose, EGTA, HEPES, KCl, $MgCl_2$, Monensin, NaCl, NaOH, $NH_3$ (32%), RbCl, Triton X-100, Trypton/Pepton and Yeast extract were purchased from Carl Roth (Graz, Austria). Agarose was obtained from VWR International (Vienna, Austria). 1,2-Bis(2-aminophenoxy)Ethane-N,N,N′,N′-Tetraacetic Acid Tetrakis Acetoxymethyl Ester (BAPTA-AM), 2-deoxy-D-glucose (2-DG), digitonin, gramicidin, nifedipine, N-methyl-D-glucamine (NMDG), Oligomycin A, propidium iodide, and tetraethylammonium chloride were obtained from Sigma Aldrich (Vienna, Austria). Bacterial protease inhibitor cocktail containing AEBSF, E-64, Bestatin, EDTA, and Pepstatin was purchased from Amresco (Cleveland, OH, USA).

Lysis buffer (in mM): 100 $Na_2HPO_4$, 200 NaCl, 10 imidazole, 250 units of Benzonase Nuclease and bacterial Protease Inhibitor Cocktail, pH = 8.0. Washing buffer (in mM) 100 $Na_2HPO_4$, 200 NaCl, 40 Imidazole, pH 8.0. Purification buffer (in mM): 100 $Na_2HPO_4$, 200 NaCl, 200 imidazole, pH 8.0. Elution buffer (in mM): 10 HEPES ±0.05% Triton X-100, pH 7.3 with NMDG. Characterization of GEPIIs in vitro was performed using elution buffer with different ions added. $EC_{50}$s in situ were determined using the following buffers containing different molarities of KCl, 10 mM HEPES and NMDG to a final molarity of 150 mM, pH adjusted to 7.4. Buffer containing 300 mM $K^+$ consisted of (in mM): 300 KCl, 10 HEPES, pH 7.4 with NMDG. Hemoglobin determination in murine sera was performed using drabkin's solution. Assay buffer for cell viability experiments was composed of (in mM) 143 NaCl, 2 $CaCl_2$, 1 $MgCl_2$, 10 HEPES either with 10 mM D-Glucose or 10 mM 2-DG. Before fluorescence microscopy experiments, cells were washed and stored for 30 min in a HEPES-buffered solution (storage buffer) containing 138 mM NaCl, 5 mM KCl, 2 mM $CaCl_2$, 1 mM $MgCl_2$, 10 mM HEPES, 2.6 mM $NaHCO_3$, 0.44 mM $KH_2PO_4$, 0.34 mM $Na_2HPO_4$, 10 mM D-glucose, 0.1% vitamins, 0.2% essential amino acids, and 1% penicillin–streptomycin, pH adjusted to 7.4 with NaOH. Buffers used for physiological measurements contained (in mM): 138 NaCl, 5 KCl, 1 $MgCl_2$, 10 D-glucose and 10 HEPES, either with 2 $CaCl_2$ ($Ca^{2+}$ buffer), 1 EGTA ($Ca^{2+}$ free buffer) or ±0.1 nifedipine or 15 tetraethylammonium chloride. Depolarization buffer was composed of (in mM): 73 NaCl, 70 KCl, 1 mM $MgCl_2$, 10 D-glucose and 10 HEPES, either with 2 $CaCl_2$ (depolarizing $Ca^{2+}$ buffer), 1 EGTA, or ±0.1 nifedipine or 15 tetraethylammonium chloride.

**Cloning of GEPIIs**. Cloning was performed according to conventional restriction digestion based procedures and all products were verified by sequencing (Eurofins Genomics, Munich, Germany). Primers used for DNA amplification (Supplementary Table 3) were purchased from Thermo Fisher Scientific (Vienna, Austria). Genomic DNA of *E. coli* 10-beta served as PCR template to isolate Kbp from bacterial cells. Single nucleotide exchanges to receive lc-BON GEPII 1.0 were performed using the primers: lc-BON 01-05 fwd and Kbp rev primer. Lc-LysM GEPII 1.0 was generated using lc-LysM 01 and 02 rev primers, and Kbp for. To obtain GEPII 2.7, GEPII 2.10 and GEPII 2.15 the primers GEPII 2.7 for/ rev, GEPII 2.10 for/ rev and GEPII 2.15 for/ rev, in combination with Kbp fwd/ rev were used. Mitochondrial targeting of GEPIIs was achieved by a tandem dimeric repeat of COX VIII targeting sequence at the N-terminal end. For nuclear targeting, the class III nuclear localization signal KRSWSMAFC was added via the primers cpV fwd and NLS cpV rev, or to observe cytosolic localization, the lysine rich motif LPPLERLTL derived from GP41 from human immunodeficiency virus was fused via the primers cpV fwd and NES cpV rev. GEPIIs were attached to the inner leaflet of the plasma membrane using the the K-ras-derived CAAX-motife MSKDVKKKKKKSKTKCVIM with CAAX cpV rev primer.

**Protein expression and purification**. Recombinant expression of GEPIIs was performed using pETM-11 bacterial expression vectors. Proteins were expressed in *E. coli* BL-21 (DE3) cells. At an $OD_{600}$ of 0.8, protein expression was induced by adding 1 mM β-D-1-thiogalactopyranoside (IPTG) and cells were incubated at room temperature. After 4 h cells were pelleted and cells were re-suspended in 20 ml of lysis buffer. Then cells were lysed by sonication and cleared by centrifugation. Proteins were purified using a 5 ml HisTrap column (GE Healthcare, Vienna, Austria) for immobilized metal affinity chromatography on an ÄKTA pure system (GE Healthcare, Vienna, Austria) at room temperature. HisTrap columns were equilibrated using lysis buffer, *E. coli* lysates were applied on the columns and washed with washing buffer. Proteins were eluted with purification buffer and the $His_6$-Protein A tag was cleaved overnight at 4 °C using 2% (w/w) of 1 mg ml$^{-1}$ recombinant His-tagged TEV protease. Processed proteins were re-purified from the fusion tags and TEV protease at room temperature using size exclusion columns (16/600 200 pg, GE Healthcare) on an ÄKTA pure system (GE Healthcare). Subsequently proteins were eluted using elution buffer.

**Characterization of purified GEPIIs**. Purified GEPIIs were analyzed using the CLARIOstar fluorescent plate reader (BMG Labtech, Ortenberg, Germany). For GEPII characterization, black CELLSTAR 96-well cell culture microplates (PS, F-Bottom, Greiner Bio-One, Kremsmünster, Austria) were used. Permanently,

GEPIIs at a final concentration of 200 nM were analyzed. Spectra were recorded using excitation at 413.4 nm ± 8 nm, No. of single recorded wavelength points: 121, Emission from 441.4 to 561.4 nm in steps of 1 nm ± 5 nm, Gain: 2000, focal height adjusted to blank. Recorded spectra were analyzed using GraphPad Prism 5 Software (GraphPad Software, Inc., La Jolla, CA, USA), equalized for area under curve and smoothed using 6th order polynomial, averaging 6 neighbors on each side. $EC_{50}$ values and specificity of GEPIIs was assessed performing single point scans using the fluorescence multichromatic mode of BMG Analysis software with the following standardized settings: Excitation 430 nm ± 10 nm, Emissions: 475 nm ± 5 nm and 525 nm ± 5 nm. Dichroic filters used: 455 and 480 nm, Gain: 2000, focal height adjusted to blank. Data was calculated dividing emissions 525/475 nm and ratio values were shown as difference from blank (0 mM [ion]). The pH stability of GEPII 1.0 was determined using Elution buffers with pH adjusted from 5.5 to 10.0. Kinetics of $K^+$ binding by GEPII 1.0 was assessed using F 4500 fluorescence spectrophotometer (Hitachi). Reversibility of $K^+$ binding by GEPII 1.0 was determined by embedding GEPII 1.0 in 0.5% agarose and performing fluorescence microscopic analysis in combination with a gravity-based perfusion system (NGFI, Graz, Austria).

**Animals.** For all experiments, male C57BL/6J mice obtained from Charles River Laboratories (Sulzfeld, Germany) were used. Mice were maintained in a clean environment with a regular light–dark cycle (12 h/12 h) and unlimited access to Chow or Western-type diet (WTD; 21% fat, 0.2% cholesterol; Ssniff Spezialdiaeten GmbH, Soest, Germany) and water. Blood was collected by either retro-bulbar or *V. facialis* puncture and serum was prepared by centrifugation for 10 min at 400 rcf (Himac CT15RE, Hitachi, Dusseldorf, Germany). To receive the bile from the animals, mice were sacrificed by cervical dislocation. Gallbladders were removed and centrifuged for 15 min at 400 rcf to release the bile (Himac). Urine was collected from mice fed WTD for 20 weeks after cervical dislocation by direct aspiration from bladders using syringe and needle. Animal experiments were carried out in accordance with the European Directive 2010/63/EU and approved by the Federal Ministry of Science, Research, and Economy, Vienna, Austria. Ischemia Reperfusion Injury was induced in anesthetized mice. Taking care of intestines, bowel and vasculature, microvascular clamps were applied to both renal pedicles for 25 min. During surgery mice were hydrated and body temperature was kept at ~37 °C using an adjustable heating pad. As a control, non-operated healthy mice were used.

**Assessment of clinical renal injury parameters.** Biochemical analysis included serum urea and creatinine (Roche Diagnostics, Mannheim, Germany). Serum neutrophil gelatinase-associated lipocalin (Lipocalin-2) was evaluated using a commercially available enzyme-linked immunosorbent assay (R&D Systems, Minneapolis, MN, USA). Measurements were performed using a FLUOstar Omega photometer (BMG Labtech, Ortenberg, Germany).

**Histology.** Formalin fixed paraffin embedded kidney tissue of healthy mice and mice after IRI was sectioned at 4 μm and stained with Periodic Acid-Schiff (Sigma-Aldrich) using a standard protocol. Finally, slides were dehydrated, cleared and mounted in anhydrous mounting medium Roti-Histokitt II (Carl Roth). Stainings were evaluated for the number of tubular casts in six high power fields (HPFs) by one blinded examiner.

**$K^+$ determination using ISE in human blood and urine samples.** A pilot study enrolling 15 patients on maintenance hemodialysis (HD) recruited from the dialysis ward of the Clinical Division of Nephrology, Medical University of Graz and 15 healthy participants without any documented kidney disease was performed. The study protocol was approved by the Internal Review Board of the Medical University of Graz (EK-Number: 29-223 ex 16/17). An informed consent was signed by all participants before enrollment. Blood and urine samples from healthy participants were collected during an ambulatory visit with blood collection syringes (BD A-Line, Plymouth, UK) and separator tubes (Greiner Bio-One, Vacuette). Both were coated with lithium-heparin. Blood samples from HD-Patients were obtained before the start of their regular hemodialysis treatment.

Spontaneous midstream urine of healthy individuals was collected in urine tubes (Greiner Bio-One, Vacuette) without any pre-analytic additives. Participants were not asked to be fasting at the time of sampling. Potassium levels within the urine and serum of healthy donors were determined using ion sensitive electrodes (Cobas 8000, Roche diagnostics, Vienna, Austria) with indirect potentiometry as test principle. Serum samples of HD patients were analyzed for $K^+$ using ion sensitive electrode (ABL 800 Flex, Drott, Vienna, Austria).

**Determination of $K^+$ within bodily fluids using GEPII 1.0.** $K^+$ levels in biological samples were assessed using a calibration curve. Measurements were performed using standard CLARIOstar setup. The equation received was solved for *x* and applied to calculate the $[K^+]$ in mMol $L^{-1}$ within biological samples, according to

the formula:

$$[K^+]\left(\frac{mMol}{L}\right) = \frac{\Delta R - 1.205}{-0.03651 - 1.205} \div (-0.002938) * \frac{Dilution}{1000}.$$

Assessment of $K^+$ levels in biological samples was performed diluting the samples with elution buffer. Additionally, murine sera were analyzed for Hemoglobin (Hb) content using drabkin's reagent and $[K^+]$ were corrected for hemolysis assuming a linear correlation between increasing Hb and increasing $[K^+]$.

**GEPII 1.0 based cell growth and cell death assay.** INS-1 832/13 (INS-1) cells were seeded one day before the experiment and showed 80–100% confluency at day of analysis. Analyses of supernatant were performed using the standard CLARIOstar protocol. Cells were permeabilized using 50 μM digitonin. Measurements at the micsoscope were performed at an iMic inverted and advanced fluorescent Microscope (TILL Photonics, Graefling, Germany). Cells were seeded the day before the experiments in 96 well μ-plates (ibidi, Munich, Germany) and analyzed at different confluences ranging from 30 to 100%. Cells were permeabilized at time point indicated by application of 30 μM digitonin. Experiments were performed using an ×20 magnification objective (alpha Plan Fluor ×20, Zeiss, Göttingen, Germany) with a motorized sample stage (TILL Photonics, Graefling, Germany). For illumination a Polychrome 5000 (TILL Photonics) was used. Emissions were recorded using a charged coupled device camera (AVT Stringray F145B, Allied Visions Technologies, Stadtroda, Germany). Filter set was obtained from AHF Analysentechnik (Tubingen, Germany). FRET-based GEPII measurements were performed at an excitation of 430 nm and emissions were collected simultaneously at 480 and 535 nm using an optical beam-splitter (Dichroic 69008-ET-ECFP/ EYFP/mCherry for CFP/YFP). Alternately, FRET and mseCFP emissions, and bright field images were acquired.

Bacterial growth experiments were performed using LB medium. $1.85*10^9$ *E. coli* cells were seeded in 39 ml at $t = 0$ and cultured for 12 h at 37 °C vigorously shaking. Directly, after 4, 8, and 12 h, and 1 additional hour at 75 °C or with 50 mg $L^{-1}$ Kanamycin, 100 mg $L^{-1}$ Ampicillin, or 50 μM Gramicidin suspensions were analyzed regarding $OD_{600}$ and 1 ml of supernatant was centrifuged at 600 rcf for 10 min (Himac) to pellet bacterial cells. Supernatant was taken, 1 vol of 30 mM NMDG containing solution was added and mixture was heated to 95 °C for 20 min to remove $NH_4^+$ from the sample and to inactivate proteases. After centrifugation at 9500 rcf for 20 min (Himac) supernatant was diluted and $[K^+]$ concentrations within the samples were calculated.

Propidium iodide (PI) staining of bacteria was performed using PI (Sigma-Aldrich) at a final concentration of 1 μg mL$^{-1}$ in PBS. PI stained cells were transferred into CELLSTAR 96-well plates PS, F-Bottom (Greiner Bio-One) and fluorimetric analysis was performed using the CLARIOstar fluorescence plate reader. PI staining solution without bacteria served as blank. Excitation was set at 535 nm ± 10 nm, emission was collected at 617 nm ± 5 nm. Dichroic filters used: 576 nm, Gain: 2000, focal height adjusted to blank.

**Two-photon microscopy.** Surgical preparation of the cremaster muscle was performed as originally described[49] in with minor modifications[50]. Mouse experiments were performed according to German legislation for the protection of animals and approved by the Regierung von Oberbayern, München, Germany.

Briefly, male mice were anesthetized with an i.p. administration of ketamine/ xylazine (100 mg kg$^{-1}$ ketamine, 10 mg kg$^{-1}$ xylazine). The right cremaster muscle was excised by a ventral incision of the scrotum, opened ventrally and spread over the pedestal of a custom-made microscopy stage. The body temperature was maintained at 37 °C using a heating pad placed under the mouse. Throughout the procedure, as well as after surgical preparation during in vivo microscopy, the muscle was superfused with warm buffered saline. In vivo imaging was performed using a Zeiss LSM 7MP microscope, equipped with a Ti:Sa laser (Chameleon Vision II, Coherent, Dieburg, Germany). Excitation wavelength was 810 nm. Light from the specimen was collected with a W-Plan-Apochromat ×20/1.0 objective (Zeiss) and detected by two LSM BIG Detectors at 460–500 and 520–560 nm. For tissue loading with GEPII 1.0, recombinant GEPII 1.0 was added into the muscle bath for 20 min (1:10 dilution) at a concentration of 4 μM. Measurements were conducted at 20, 70, and 130 μm depth in the muscle tissue at the indicated time points.

**Cell culture and transfection.** DMEM (Sigma Aldrich) containing 10% fetal bovine serum, 100 U ml$^{-1}$ penicillin, 100 μg gn$^{-1}$ streptomycin, and 2.5 μg ml$^{-1}$ Fungizone (Thermo Fisher Scientific) was used to grow EA.hy926, HeLa and HEK293a cells (all obtained from ATCC, Guernsey, UK). INS-1 cells were obtained from C.B. Newgard, Department of Pharmacology and Cancer Biology, Duke University School of Medicine, USA. INS-1 cells were cultivated in GIBCO RPMI Medium 1640 obtained from Thermo Fisher Scientific, additionally supplemented with 10% FCS, 10 mM HEPES, 1 mM Sodium pyruvate, 0.05 mM 2-mercaptoethanl, 100 U ml$^{-1}$ penicillin, 100 μg gn$^{-1}$ streptomycin and 2.5 μg ml$^{-1}$ Fungizone (Thermo Fisher Scientific). Transfection of cells in 30-mm imaging dishes was performed using TransFast transfection reagent (Promega GmbH,

Mannheim, Germany)[51]. To load cells with BAPTA-AM, culture media was removed from cells and replaced with 37 °C warm nominally $Ca^{2+}$ free buffer containing 50 µM of BAPTA-AM directly before fluorescence microscopic experiments. Cells were stored for 45 min in a humidified incubator (37 °C, 5% $CO_2$). Excessive BAPTA-AM was removed by washing steps with 37 °C warm nominally $Ca^{2+}$ free buffer.

**Live cell imaging**. Prior to the measurements cells were equilibrated in storage buffer for 30 min. During the experiment, buffers were exchanged using a flow chamber, connected to a gravity-based perfusion system (NGFI, Graz, Austria) and a vacuum pump (Chemistry diaphragm pump ME 1c, Vacuubrand, Wertheim, Germany). Fluorescence microscopic experiments were performed at an iMic inverted and advanced fluorescent Microscope using an ×40 magnification objective (alpha Plan Fluor ×40, Zeiss, Göttingen, Germany) with a motorized sample stage (TILL Photonics, Graefling, Germany)[21]. $Ca^{2+}$ imaging using Car-GECO1 was performed at an excitation of 575 nm and emission was collected at 600 nm. For control and acquisition the software Live acquisition 2 (TILL Photonics) was used. Subcellular targeting of GEPIIs was ensured using an array confocal laser scanning microscope (ACLSM)[31]. Targeted GEPIIs were illuminated using 488 nm laser light and emission was collected at 535 nm at a binning of 2 using a CCD camera (CoolSnap HQ2, Photometrics, Tucson, Arizona, USA).

**Data analysis**. Analysis of raw data acquired was performed using excel (Microsoft, Washington, US) in combination with GraphPad Prism5 software (GraphPad Software). Bleaching correction of live cell imaging data was performed by curve fitting. Representative ratio images were created using MetaMorph microscopy automation and image analysis software (Molecular Devices, Sunnyvale, CA, USA). Background correction was performed and FRET and mseCFP images were divided with appropriate set ratio range. Statistical analysis of data was performed using GraphPad Prism5 software, either performing paired *t*-test, unpaired *t*-test or one-way ANOVA test with Tukey's Multiple Comparison Test as indicated in the figure legends.

**Data availability**. Original data are available from the corresponding author upon reasonable request.

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

## Acknowledgements

The authors acknowledge C.J. Edgell, Pathology Department, University of North Carolina at Chapel Hill, NC, USA for providing the EA.hy926 cells. We thank C.B. Newgard, Department of Pharmacology and Cancer Biology, Duke University School of Medicine, USA, for providing us with INS-1 832/13 cells. CMV-CAR-GECO1 was a kind gift from Robert Campbell, Department of Chemistry, University of Alberta, Edmonton, Alberta, Canada (Addgene plasmid # 45493). We thank Karin Osibow for editing the manuscript. The authors further acknowledge the scientific advisory board of Next Generation Fluorescence Imaging (NGFI) GmbH (http://www.ngfi.eu/), a spin-off company of the Medical University of Graz. The research was funded by the Ph.D. program Molecular Medicine (MOLMED) of the Medical University of Graz, by Nikon Austria within the Nikon-Center of Excellence, Graz, the FWF projects P28529-B27, the doctoral program Metabolic and Cardiovascular Disease (DK-W1226), and P27070, and the BioTechMed-Graz flagship project Lipases and Lipid Signaling, and Molecular Fundamentals of Inflammation (DK-MOLIN, W1241) of the Medical University of Graz. The Nikon Center of Excellence, Graz, is supported by the Austrian infrastructure program 2013/2014, Nikon Austria Inc., and BioTechMed, Graz.

## Author contributions

H.B. designed and generated GEPIIs, performed experiments, analyzed data and wrote the manuscript. M.R. and N.P. designed, performed and analyzed in vivo imaging experiments. S.S. together with T.M. purified recombinant GEPIIs; K.A., K.E., A.R.R., N.V., and D.K. designed, performed and analyzed animal experiments; M.W.-W. and E.E. contributed to experiments and data analysis; B.G. established and performed analysis; R.R. performed cell culture experiments and transfections; A.T.D. and T.N. collected human samples, performed and analyzed respective experiments; H.L. and R.P. assisted in FRET imaging of purified GEPIIs; B.P. and K.G. contributed to experimental design and data interpretation. W.F.G. together with R.M. designed and supervised the project, and wrote the manuscript.

## Additional information

**Competing interests:** Together with the legal representatives of the Medical University of Graz (https://www.medunigraz.at/en/) the inventors of GEPIIs and authors of this paper, H.B., E.E., W.F.G., R.M., and M.W.-W. have filed a patent application at the Austrian patent office (https://www.patentamt.at/en/) with the application number A50400/2017. Purified GEPIIs and plasmids coding for GEPIIs mentioned in this manuscript are commercially available upon request at Next Generation Fluorescence Imaging (NGFI) GmbH (http://www.ngfi.eu/), a private spin-off company of the Medical University of Graz. E.E., W.F.G., R.M., and M.W.-W. are shareholders of NGFI. The remaining authors declare no competing financial interests.

