## [Peer Review File · Nature Communications]

Reviewers' comments:

Reviewer #1 (Remarks to the Author):

The manuscript by Malli and coworkers reports the successful development and characterization of the first K⁺ specific FRET-based sensor protein. This represents an important achievement both from the application perspective and from a protein engineering perspective. The development of specific metal ion sensors for relatively weak binding metal ions such as Mg²⁺, but in particular K⁺ and Na⁺, is inherently challenging. The sensor proteins reported here are remarkably robust, both in terms of metal selectivity, thermodynamic stability and the change in emission ratio. The success of this FRET sensor is based on their use of the recently characterized bacterial K⁺ binding protein Kbp, which combines high thermodynamic stability and metal selectivity with a large metal-induced conformational change that is ideally suited for use in FRET-based sensing. Overall I was impressed by the careful characterization of sensor properties and the demonstration of its potential applications, ranging from in vitro diagnostics to intracellular imaging. I therefore recommend publication pending minor revision to address the following questions/remarks.

Comments/questions:

- Only a schematic design of the various fusion proteins is reported. It would be useful to discuss the design considerations in a little more detail (choice of linkers, choice of FPs), and also include in the supporting information the exact amino acid sequence of the various reporters. In addition, I wondered whether the authors tried to make variants in which only a single E or D residue was mutated, potentially yielding variants with different K⁺ affinities.
- It would be insightful to discuss the origin of the remarkable large change in emission ratio, which is both a result of the surprisingly low FRET in the K⁺-free state and remarkable efficient FRET in the K⁺-bound state. Do the fluorescent domains that are used have a tendency to dimerize?
- p10. 'As found previously for Ca²⁺ sensors' suggests that this is a general property of Ca²⁺ sensors, which is not true. In addition, the in situ determined EC₅₀ values appear not too much different from the K_d values obtained in vitro.
- I was very surprised by the higher K⁺ concentrations in the nucleus reported here compared to the cytosol. As stated by the authors, there has been little evidence that indicates the existence of concentration gradients between the nucleus and cytosol for K⁺ or other metal ions. The reason that this is surprising, is that these small ions are generally assumed to be able to freely diffuse through the nuclear pore. The concentration determination appears to be done by comparing the absolute emission ratio observed for nuclear targeted sensors to that of cytosolic sensors. However, this may be dangerous since background fluorescence, which could be different between organelles, also contributes to the emission ratio. Sensors with a high emission ratio would be particularly sensitive to such artefacts, as a relatively small amount of background fluorescence in the CFP channel will result in a relatively large difference in emission ratio. In this respect the authors might want to be a little bit more conservative in their claims.
- the method of proving reversible binding reported here is somewhat indirect, as it looks at the effect of temperature. Did the authors consider the use of chelators/crown ethers to deplete the K⁺ concentration?

Reviewer #2 (Remarks to the Author):

The manuscript by Bischof and colleagues presents a new genetically encoded FRET sensor to monitor potassium concentrations. It is based on a recently described bacterial potassium binding protein that undergoes a conformational change when binding potassium, thereby increasing FRET efficiency of suitable fluorescent proteins fused to it. Several affinity variants of the sensor are presented. It was shown to work in determining potassium concentrations in several body fluids and for ratiometric imaging of potassium in live cells. There are currently only limited ways to

monitor potassium in cells, mainly synthetic fluorescent potassium sensors such as PBFI that are not specific enough and cannot be targeted by genetic means. Thus, the sensor will be useful.

I have a few issues that should be addressed before I can recommend publication.

-How many potassium ions bind a sensor molecule? Please provide data on potential cooperativity and indicate Hill co-efficients.

-there are currently no data on the kinetics of potassium binding to and release from the sensor.

This is important to know for any time-resolved live cell imaging experiments

-Were there any attempts made to increase FRET change of the sensor by applying molecular engineering?

- Figure 5 and discussion: I find it hard to believe that nuclear potassium levels should be higher than cytosolic levels. The nuclear pores are large enough for small proteins to diffuse through them. Why should they pose an obstacle for potassium ions? There had been a similar discussion on nuclear calcium levels being different from cytosolic ones, but that was an artefact. Later it was convincingly shown that cytosolic calcium waves simply pass through the nucleus, without any noticeable delay.

Reviewer #3 (Remarks to the Author):

This paper describes the development and use of a series of K⁺-sensitive FRET probes, that reliably record the concentration of K⁺ in vitro, in living cells and in vivo. Unlike other ions, our understanding of the regulation and spatio-temporal dynamics of K⁺ has been severely hampered by the lack of specific, non-toxic and kinetically fast probes that can operate over a range of K⁺ concentrations seen within and outside living cells. This study reports the design of GEPIIs and use them for quantitative real-time imaging of K⁺ dynamics. The authors nicely develop GEPIIs with a range of K⁺ affinities. The experiments are well done, carefully controlled and contain important new information. I have the following comments.

1. The authors nicely show the selectivity of GEPIIs for different monovalent cations. However, these experiments were done in the presence of each ion alone. Physiologically, there will be Na⁺ and K⁺ present. The authors should therefore check whether the FRET signal to external K⁺ concentrations (typically 4-5 mM) are affected by the presence of physiological external Na⁺ of ~135 mM.

2. It is very surprising that nuclear K⁺ is ~3.5 fold > cytosolic K⁺. This means nuclear K⁺ is around 420 mM, which seems very high. It is difficult to see how such a standing gradient could be maintained; the nuclear pores provide a low resistance diffusion pathway into and out of the nucleus and calcium ions, for example, diffuse freely from the cytosol into the nucleus. How is this K⁺ gradient maintained? Alternatively, could the viscosity of the nucleoplasm or other factors be artificially distorting the FRET signal, giving this very high estimate?

3. The data in Figure 6 could be strengthened by using a calcium-sensitive K channel blocker, rather than TEA.

4. Another example of a K⁺-activated enzyme in addition to pyruvate kinase should be cited; the text refers to many such enzymes but only one is mentioned and with 2 references.

Point-by-Point Response Letter

NCOMMS-17-17151-T: *“Novel genetically encoded fluorescent probes enable real-time detection of potassium in vitro and in vivo”*

Reviewer #1 (Remarks to the Author):

The manuscript by Malli and coworkers reports the successful development and characterization of the first K⁺ specific FRET-based sensor protein. This represents an important achievement both from the application perspective and from a protein engineering perspective. The development of specific metal ion sensors for relatively weak binding metal ions such as Mg²⁺, but in particular K⁺ and Na⁺, is inherently challenging. The sensor proteins reported here are remarkably robust, both in terms of metal selectivity, thermodynamic stability and the change in emission ratio. The success of this FRET sensor is based on their use of the recently characterized bacterial K⁺ binding protein Kbp, which combines high thermodynamic stability and metal selectivity with a large metal-induced conformational change that is ideally suited for use in FRET-based sensing. Overall I was impressed by the careful characterization of sensor properties and the demonstration of its potential applications, ranging from in vitro diagnostics to intracellular imaging. I therefore recommend publication pending minor revision to address the following questions/remarks.

ANSWER: We highly appreciate the very positive evaluation of our manuscript by reviewer 1. Moreover, we thank reviewer 1 for the insightful remarks, which helped us to further improve the quality of our work.

Comments/questions:

- Only a schematic design of the various fusion proteins is reported. It would be useful to discuss the design considerations in a little more detail (choice of linkers, choice of FPs), and also include in the supporting information the exact amino acid sequence of the various reporters. In addition, I wondered whether the authors tried to make variants in which only a single E or D residue was mutated, potentially yielding variants with different K⁺ affinities.

ANSWER: In order to discuss the design consideration in a little more detail, we have revised the text on page 2 lines 33-35, page 3 line 1 and 21-23, and page 9 lines 16-24. In addition, we have prepared new Supplementary Fig. 2, in which we provide the full amino acid sequences of the different Kbp variants of all constructs as suggested by reviewer 1. So far, we have not tested single E or D mutations. However, we agree that this might lead to novel GEPIIs with different K⁺ affinities. We have mentioned this possibility in the revised manuscript on page 9 lines 20-21.

- It would be insightful to discuss the origin of the remarkable large change in emission ratio, which is both a result of the surprisingly low FRET in the K⁺-free state and remarkable efficient FRET in the K⁺-bound state. Do the fluorescent domains that are used have a tendency to dimerize?

ANSWER: We have discussed the putative reason for the large FRET ratio changes in the revised version of our manuscript. As written on page 11, lines 11-13, we feel certain that a dramatic conformational change of Kbp from its elongated (K⁺ unbound) to its spherical (K⁺ bound) state is responsible for the huge FRET ratio changes. Such a rearrangement has been described by Ashraf et al. (Structure 2016; PMID:27112601). To our knowledge, the used FPs have a rather weak tendency to dimerize. This was mentioned in the revised manuscript on page 3 line 1.

- p10. 'As found previously for Ca²⁺ sensors' suggests that this is a general property of Ca²⁺ sensors, which is not true. In addition, the in situ determined EC50 values appear not too much different from the K_d values obtained in vitro.

ANSWER: We thank reviewer1 for this important clarification. We have removed the respective statement from the revised version of the manuscript.

- I was very surprised by the higher K⁺ concentrations in the nucleus reported here compared to the cytosol. As stated by the authors, there has been little evidence that indicates the existence of concentration gradients between the nucleus and cytosol for K⁺ or other metal ions. The reason that this is surprising, is that these small ions are generally assumed to be able to freely diffuse through the nuclear pore. The concentration determination appears to be done by comparing the absolute emission ratio observed for nuclear targeted sensors to that of cytosolic sensors. However, this may be dangerous since background fluorescence, which could be different between organelles, also contributes to the emission ratio. Sensors with a high emission ratio would be particularly sensitive to such artefacts, as a relatively small amount of background fluorescence in the CFP channel will result in a relatively large difference in emission ratio. In this respect the authors might want to be a little bit more conservative in their claims.

ANSWER: We were also surprised by the finding that K⁺ within the nucleus seems to be much higher than in the rest of the cell. Initially, we also assumed that this might be due to an artefact as suggested by the reviewer. However, having used three different nucleus targeted constructs with the same FRET pairs but different K⁺ affinities have clearly supported our claim. From our experience, these experiments rather exclude any artefacts related to nucleus targeting. We feel that the unexpected high nuclear K⁺ concentration, first time reported in an intact living cell by the usage of GEPIIs, is for sure a provocative finding, which should, however, be published; particularly, as it has the potency to stimulate further

important research in this regard and might have multiple implications in cell physiology and pathology. However, we agree to somehow weaken our claims in this regard. Hence, we have carefully discussed this important aspect in the revised version of our manuscript. Please see page 8 lines 5-6, and page 10 lines 11-15 and 20-26.

- the method of proving reversible binding reported here is somewhat indirect, as it looks at the effect of temperature. Did the authors consider the use of chelators/crown ethers to deplete the K⁺ concentration?

ANSWER: We agree that the experiments with recombinant GEPII at different temperature if at all, only indirectly point to the reversibility of the K⁺ probe. Hence, we removed the respective statement. To further demonstrate that recombinant GEPIIs also work in a reversible manner we imaged GEPIIs embedded in agarose matrix on an inverted fluorescence microscope that was equipped with a gravity driven perfusion system. This allowed us to dynamically manipulate the K⁺ concentration fast and report the respective FRET ratio changes of purified GEPII in vitro. The new data sets are shown in new Supplementary Fig. 3f and described on page 3 lines 29 – 32 of the revised manuscript.

As useful K⁺ chelators/crown ethers have not been available in our laboratory, we could not perform meaningful experiments in this regard. However, we feel that the new experiments with recombinant GEPII 1.0 in agarose matrix as well as experiments with GEPIIs expressed in cells (Fig. 5b) convincingly demonstrate that the novel genetically encoded K⁺ probes respond to K⁺ in a fully reversible manner.

Reviewer #2 (Remarks to the Author):

The manuscript by Bischof and colleagues presents a new genetically encoded FRET sensor to monitor potassium concentrations. It is based on a recently described bacterial potassium binding protein that undergoes a conformational change when binding potassium, thereby increasing FRET efficiency of suitable fluorescent proteins fused to it. Several affinity variants of the sensor are presented. It was shown to work in determining potassium concentrations in several body fluids and for ratiometric imaging of potassium in live cells. There are currently only limited ways to monitor potassium in cells, mainly synthetic fluorescent potassium sensors such as PBFI that are not specific enough and cannot be targeted by genetic means. Thus, the sensor will be useful.

I have a few issues that should be addressed before I can recommend publication.

ANSWER: We thank reviewer 2 for the positive evaluation of our manuscript and important comments that helped us to further improve the quality of our work.

-How many potassium ions bind a sensor molecule? Please provide data on potential cooperativity and indicate Hill co-efficients.

ANSWER: We agree that for complete sensor characterization the binding properties should be analyzed. Accordingly, we have performed additional analysis to show that K^+ binding to GEPIIs happens in a non-cooperative manner. Respective Hill slopes are close to 1. The new data are shown in new Supplementary Fig. 3a-e and described on page 3 lines 27-29. We also mentioned that the binding properties found confirm the original study of Ashraf et. al (Structure 2016; PMID:27112601), indicating that one molecule Kbp binds one K^+ ion.

-there are currently no data on the kinetics of potassium binding to and release from the sensor. This is important to know for any time-resolved live cell imaging experiments

ANSWER: We agree that the association rate constant (k_{on}) and dissociation rate constant (k_{off}) are important parameters. Hence, we have performed additional experiments and provide the new data in new Supplementary Fig. 1a. Respective results are described on page 3 lines 13-15.

-Were there any attempts made to increase FRET change of the sensor by applying molecular engineering?

ANSWER: We have not yet performed any systematic molecular engineering approaches to improve the existing GEPIIs. The available GEPIIs mentioned in the manuscript were all rationally designed and generated using classical cloning strategies. However, in the revised version of our manuscript we have discussed future development strategies, which have the potency to further improve GEPIIs. In this regard we cited a landmark publication (Zhao et. Al Science 2011), in which the potency of molecular engineering to come up with a palette of improved genetically encoded Ca^{2+} probes was impressively demonstrated. This should be also possible for the novel GEPIIs in future. The discussion can be found on page 9 lines 22-24.

- Figure 5 and discussion: I find it hard to believe that nuclear potassium levels should be higher than cytosolic levels. The nuclear pores are large enough for small proteins to diffuse through them. Why should they pose an obstacle for potassium ions? There had been a similar discussion on nuclear calcium levels being different from cytosolic ones, but that was an artefact. Later it was convincingly shown that cytosolic calcium waves simply passage through the nucleus, without any noticeable delay.

ANSWER: We agree that the high K^+ concentration within the nucleus is surprising, but feel that our data are correct and not due to an artefact. It is feasible that the diffusion of K^+ within the nucleus is limited e.g. by binding of K^+ to DNA or other components. This would

hinder K^+ ions to freely pass the nuclear pores and allow Na^+/K^+ pumps to build up a K^+ gradient between the nucleus and cytosol while pumping K^+ ions from the nuclear envelope into the nucleoplasm. However, as also suggested by reviewer 1 we have weakened our claims in this regard, while we decided to publish this potentially highly relevant and stimulating finding in this manuscript. The manuscript was revised on page 8 lines 5-6, and page 10 lines 11-15 and 20-26 in this regard.

Reviewer #3 (Remarks to the Author):

This paper describes the development and use of a series of K^+ -sensitive FRET probes, that reliably record the concentration of K^+ in vitro, in living cells and in vivo. Unlike other ions, our understanding of the regulation and spatio-temporal dynamics of K^+ has been severely hampered by the lack of specific, non-toxic and kinetically fast probes that can operate over a range of K^+ concentrations seen within and outside living cells. This study reports the design of GEPIIs and use them for quantitative real-time imaging of K^+ dynamics. The authors nicely develop GEPIIs with a range of K^+ affinities. The experiments are well done, carefully controlled and contain important new information. I have the following comments.

ANSWER: We thank reviewer 3 for the very positive evaluation of our manuscript and important comments.

1. The authors nicely show the selectivity of GEPIIs for different monovalent cations. However, these experiments were done in the presence of each ion alone. Physiologically, there will be Na^+ and K^+ present. The authors should therefore check whether the FRET signal to external K^+ concentrations (typically 4-5 mM) are affected by the presence of physiological external Na^+ of ~135 mM.

ANSWER: We thank reviewer 3 for this insightful comment and performed respective experiments. The new data are presented in new Supplementary Fig. 10 and described in the main text on page 7 lines 25-29. As shown in the revised version of our manuscript even high Na^+ levels/fluctuations did not significantly influence the properties of GEPII 1.0 to sense K^+ .

2. It is very surprising that nuclear K^+ is ~3.5 fold > cytosolic K^+ . This means nuclear K^+ is around 420 mM, which seems very high. It is difficult to see how such a standing gradient could be maintained; the nuclear pores provide a low resistance diffusion pathway into and out of the nucleus and calcium ions, for example, diffuse freely from the cytosol into the nucleus. How is this K^+ gradient maintained? Alternatively, could the viscosity of the

nucleoplasm or other factors be artificially distorting the FRET signal, giving this very high estimate?

ANSWER: We agree that this finding is unexpected. However, we feel that our data are correct in this regard, particularly as we used three different nuclear targeted GEPIIs with different K^+ affinities. These experiments rather exclude any artefacts related to nuclear targeting. However as also suggested by reviewer 1, we weakened the claims and further discussed this important aspect. The respective changes can be found on page 8 lines 5-6, and page 10 lines 11-15 and 20-26 within the revised manuscript.

3. The data in Figure 6 could be strengthened by using a calcium-sensitive K channel blocker, rather than TEA.

ANSWER: We agree that the usage of specific K_{Ca} channel inhibitors would basically strengthen this set of experiments. However, the pancreatic beta cells used might express different types of K_{Ca} channels. This assumption is supported by several publications: PMID: 12882916, PMID: 19401418, PMID: 18390794, and PMID: 12830383. Hence, we would need to use different selective inhibitors such as iberitoxin or charybdotoxin for the big BK_{Ca} , apamin for the small conductance K_{Ca} channel and possibly additional ones for the intermediate K_{Ca} channels. Overall, the result of such, in a constant flow experiment exorbitantly expensive mixture, would be the same that we achieved with TEA. While we agree that the suggested experiments are meaningful to study the individual contribution of each type of calcium-activated K^+ channel to subcellular K^+ movements, we believe that such study is clearly beyond the focus of the current manuscript, in which we introduce the novel genetically encoded K^+ probes for the very first time. We hope the referee kindly agrees with our decision not to perform additional experiments in this regard.

4. Another example of a K^+ -activated enzyme in addition to pyruvate kinase should be cited; the text refers to many such enzymes but only one is mentioned and with 2 references.

ANSWER: In the revised version of our manuscript we have mentioned additional enzymes which are known to be regulated by K^+ . Please see page 2 lines 8-9.

REVIEWERS' COMMENTS:

Reviewer #1 (Remarks to the Author):

The authors addressed all my questions and suggestions in a satisfactory manner. As the other 2 reviewers I remain somewhat skeptical on the perceived differences in K⁺ concentration between the nucleus and the cytosol. The authors argument that they observed this using 3 different sensors, could also argue against their hypothesis, as one would not expect to see differences in the emission ratio for the low affinity LC-BON GEPII 1.0 sensor or the high affinity GEPII 1.0 sensors. Also, in my opinion their results cannot be explained by the presence of a high concentration of K⁺ binding proteins in the nucleus, as these would not affect the amount of free K⁺ ions when K⁺ ions could diffuse freely through the nuclear pore. In order to have a gradient, diffusion should be restricted in some way and/or specific K⁺ pumps should be present. Nonetheless, since the authors are aware of and openly discuss the provocative nature of their findings, I agree that there is also value in giving other scientist the opportunity to think about these findings. I therefore support publication of this beautiful work.

Reviewer #2 (Remarks to the Author):

The authors have answered my questions in a satisfying manner.

Reviewer #3 (Remarks to the Author):

The authors have satisfactorily addressed my concerns. Whilst I remain sceptical that nuclear K⁺ concentration can be 3-4 times higher than cytosolic levels, the authors have ruled out possible sources of error. As they state themselves, this is a provocative finding and will certainly spur further activity in this important but neglected area.

Point-by-Point Response Letter

NCOMMS-17-17151-A: *“Novel genetically encoded fluorescent probes enable real-time detection of potassium in vitro and in vivo”*

We thank the referees very much for their kind comments and their valuable help for improving our work. We have addressed the remaining minor issue as indicated below.

Reviewer #1 (Remarks to the Author):

The authors addressed all my questions and suggestions in a satisfactory manner. As the other 2 reviewers I remain somewhat skeptical on the perceived differences in K⁺ concentration between the nucleus and the cytosol. The authors argument that they observed this using 3 different sensors, could also argue against their hypothesis, as one would not expect to see differences in the emission ratio for the low affinity LC-BON GEPII 1.0 sensor or the high affinity GEPII 1.0 sensors. Also, in my opinion their results cannot be explained by the presence of a high concentration of K⁺ binding proteins in the nucleus, as these would not affect the amount of free K⁺ ions when K⁺ ions could diffuse freely through the nuclear pore. In order to have a gradient, diffusion should be restricted in some way and/or specific K⁺ pumps should be present. Nonetheless, since the authors are aware of and openly discuss the provocative nature of their findings, I agree that there is also value in giving other scientist the opportunity to think about these findings. I therefore support publication of this beautiful work.

ANSWER: We highly appreciate the positive evaluation and support of reviewer 1.

We understand the review's argument regarding the usage of the 3 different K⁺ probes, while in line with our argumentation the greatest differences were obtained with the Ic-LysM GEPII 1.0, the sensor with the most appropriate K_d (~ 60 nM, See Suppl. Fig. 11 c).

We also agree that a simple K⁺ buffer system within the nucleoplasm would not explain the existence of a respective K⁺ gradient, assuming free diffusion of K⁺ through nuclear pores. As suggested by reviewer 1 other mechanisms such as K⁺ pumps and/or a limited diffusion of K⁺ across the nuclear envelope might contribute to respective K⁺ gradients. Indeed, the presence of functional Na⁺/K⁺ ATPases within the nuclear envelope has been described (see e.g. M.H. Graner; The Journal of Membrane Biology 2002; PMID: 12029368). Consequently, in the final version of our manuscript we have revised the respective discussion about mechanisms that might allow generating a K⁺ gradient between the nucleus and cytosol. Particularly, we removed the statement about K⁺ buffering within the nucleus as a mechanism that facilitate the generation of a respective K⁺ gradient.

Now it is written: *“However, organelle targeting of fluorescent probes can cause artefacts due to differences in auto fluorescence, viscosity and composition, which might produce false results. To exclude any of these disturbances, we tested three different GEPIIs with same FPs but different K⁺*

affinities within the different cellular compartments. Nevertheless the high $[K^+]_{nuc}$ is surprising, as until now there has been little evidence on the distribution of K^+ within organelles, apart from studies from the 1970s⁴⁰ and 1980s⁴¹ that indicated the existence of K^+ gradients between the cytosol and nucleus. K^+ within the nucleus has been suggested to contribute to the control of gene expression⁴¹, but although some K^+ channels and transporters are known to be present in the nuclear envelope^{13,42}, the mechanisms that might dynamically regulate $[K^+]_{nuc}$ ⁴² and $[K^+]_{mito}$ ^{42,43} are not well understood. Although nuclear pores allow the passage of macromolecules⁴⁴, their permeability for K^+ ions remains elusive. ~~Even if nuclear pores are freely permeable for K^+ ions, the diffusion of K^+ from the nucleus to the cytosol might be limited by high concentrations of K^+ -binding domains within the nucleoplasm. Several studies⁴⁵⁻⁴⁷ demonstrated that certain DNA structures specifically bind K^+ with high affinity, indicating a certain role of K^+ within the nucleus.~~ However, further experiments are necessary to validate any peculiarities of the K^+ homeostasis of cellular nuclei.”

Reviewer #2 (Remarks to the Author):

The authors have answered my questions in a satisfying manner.

ANSWER: We once again would like to thank reviewer 2 for the positive evaluation of our manuscript and important comments that helped us to further improve the quality of our work.

Reviewer #3 (Remarks to the Author):

The authors have satisfactorily addressed my concerns. Whilst I remain sceptical that nuclear K^+ concentration can be 3-4 times higher than cytosolic levels, the authors have ruled out possible sources of error. As they state themselves, this is a provocative finding and will certainly spur further activity in this important but neglected area.

ANSWER: We thank reviewer 3 for the very positive evaluation of our manuscript as well as important, critical and insightful comments.